# Immunobiology of Thymic Epithelial Tumors: Implications for Immunotherapy with Immune Checkpoint Inhibitors

**DOI:** 10.3390/ijms21239056

**Published:** 2020-11-28

**Authors:** Valentina Tateo, Lisa Manuzzi, Andrea De Giglio, Claudia Parisi, Giuseppe Lamberti, Davide Campana, Maria Abbondanza Pantaleo

**Affiliations:** 1Department of Experimental, Diagnostic and Specialty Medicine, Policlinico di Sant’Orsola University Hospital, Via P. Albertoni 15, 40138 Bologna, Italy; valentina.tateo@studio.unibo.it (V.T.); lisa.manuzzi@studio.unibo.it (L.M.); andrea.degiglio@studio.unibo.it (A.D.G.); claudia.parisi4@studio.unibo.it (C.P.); 2Oncologia Medica, Azienda Ospedaliero-Universitaria di Bologna, Via Albertoni 15, 40138 Bologna, Italy; davide.campana@unibo.it (D.C.); maria.pantaleo@unibo.it (M.A.P.)

**Keywords:** thymic epithelial tumors, thymoma, thymic carcinoma, immunobiology, autoimmunity, immuno-oncology, immune check-point inhibitors, biomarkers, PD-L1, immune-related adverse events

## Abstract

Thymic epithelial tumors (TETs) are a group of rare thoracic malignancies, including thymic carcinomas (TC) and thymomas (Tm). Autoimmune paraneoplastic diseases are often observed in TETs, especially Tms. To date, chemotherapy is still the standard treatment for advanced disease. Unfortunately, few therapeutic options are available for relapsed/refractory TETs. In the last few years, the deepening of knowledge on thymus’ immunobiology and involved altered genetic pathways have laid the foundation for new treatment options in these rare neoplasms. Recently, the immunotherapy revolution has landed in TETs, showing both a dark and light side. Indeed, despite the survival benefit, the occurrence of severe autoimmune treatment-related adverse events has risen crescent uncertainty about the feasibility of immunotherapy in these patients, prone to autoimmunity for their cancer biology. In this review, after summarizing immunobiology and immunopathology of TETs, we discuss available data on immune-checkpoint inhibitors and future perspectives of this therapeutic strategy.

## 1. Introduction

Thymic epithelial tumors (TETs) are a group of rare thoracic malignancies including thymic carcinomas (TC) and thymomas (Tm), with an annual incidence rate ranging between 0.9 and 2.3 per million [1]. Tms are further classified into type A, atypical type A, type AB, B1, B2 and B3, depending on the histopathological feature of epithelial tumor cells and the proportion of the lymphocytic component [2,3]. While TC is an aggressive disease with high metastatic potential, Tms are indolent slow-growing tumors, but are often associated with autoimmune paraneoplastic syndromes, such as myasthenia gravis (MG) [4,5].

Upfront surgery is the main treatment strategy for TETs: the achievement of complete resection (R0) together with tumor stage at diagnosis is the most important prognostic factors at baseline. In fact, the 5-year overall survival after radical surgery is 90% for stages I and II, 60% in stage III and less than 25% for stage IV TETs [4,6,7,8]. Post-operative treatment with adjuvant intent (either chemotherapy, radiotherapy or concomitant chemoradiotherapy) can be considered on an individual basis depending on stage, margin infiltration and histology subtype [4,6,7,9,10].

Unresectable TETs should undergo systemic polychemotherapy, but no randomized clinical trial has clarified which regimen should be considered as the standard of care [4,6,7]. Polychemotherapy with cisplatin, doxorubicin and cyclophosphamide is the first choice for Tms, yielding an overall response rate (ORR) of 50% and a median duration of response (mDOR) of 11.8 months [11], while the combination of carboplatin and paclitaxel is preferred for TCs, and yields an ORR of 21.7% and a median progression-free survival (mPFS) of 5 months [12]. The second and subsequent lines of treatment need expert evaluation, since available data are derived from non-randomized prospective and retrospective studies as well as case reports [4,6,7].

TET patients often present with immune-mediated paraneoplastic syndromes. The role of thymus in the development of adaptive immunity and the altered molecular pathways involved in TET might explain the high rate of paraneoplastic syndromes in these patients [13,14,15,16]. Immunotherapy with immune checkpoint inhibitors is being investigated in TETs, yielding interesting response rates and survival outcomes. However, severe toxicity has been observed in TET patients secondary to abnormal immune responses promoted by immune checkpoint inhibition, thus limiting their routine application in clinical practice. We reviewed the most recent understanding of TET immunopathology, and existing data on immunotherapy to put into frame the immune-related events in TET patients and provide a focus for ongoing studies and future perspectives in the treatment of this rare group of diseases.

## 2. Immunobiology of Thymus

Thymus, together with bone marrow, is a primary lymphoid organ and is responsible for T-cell development. An outer capsule, a cortex and a medulla can be recognized in the thymus structure.

Thymic epithelial cells derive from the pharyngeal endoderm and are differentially organized in a loose net in the cortex, while they cluster in solid cordons in the medulla, as well as structures known as Hassall corpuscles. Lymphocytes, named “T-cells” after thymus, derive from thymocytes and are intermingled with epithelial cells [17]. T-cell progenitors move through the thymic structure and undergo several transformations as they differentiate in the thymus, before becoming mature T-cells (Figure 1). The most immature CD3+CD4-CD8- (double negatives, DN) thymocytes can differentiate in the cortex into CD3+CD4+CD8+ (double positive, DP) cells only after a process called β selection, that consists of rearrangement and expression of functional T-cell receptors (TCR) β chains [17,18]. DP thymocytes then rearrange the TCR α chain genes and express TCR-α/β, and undergo positive and negative selection. By positive selection, only those thymocytes that have a TCR capable of binding the Major Histocompatibility Complex (MHC) expressed on thymic epithelial cells are preserved. CD8 functions as a coreceptor to bind to MHC class I, while CD4 binds MHC class II. Only positively selected thymocytes polarize towards a persistent expression of either CD4 or CD8, depending on their specificity, and can mature into either CD4+ or CD8+ single positive (SP) T-cells. SP cells migrate in the medulla, where negative selection takes place: all autoreactive thymocytes binding with a high affinity to self-peptides presented on MHC I and MHC II are eliminated by apoptosis [17,18].

Auto Immune Regulator (AIRE) is a transcriptional factor highly expressed in medullary thymic epithelial cells (mTECs), which plays a pivotal role in T-cell negative selection. Indeed, it promotes the ectopic transcriptional activity of a large number of chromosomal locations, which in turn favors the expression of tissue-restricted antigens (TRAs) selectively expressed in specific tissues [19]. On the other hand, as some TRAs are also expressed by tumors, TRAs expression in mTECs, determines a cancer-antigen specific tolerance, leading to a reduced antitumor immune response [20]. Recently, the forebrain embryonic zinc finger-like protein 2 (Fezf2) has been described as a new key transcription factor in the negative selection process. Fezf2 is selectively expressed in thymic medullary epithelial cells, it regulates a large number of TRAs and suppresses the onset of an autoimmune response. Fezf2 and AIRE regulate the expression of different TRAs with only partial overlap [21].

Nonetheless, negative selection is not completely efficient, since some autoreactive T-cells could escape thymic selection and be released into the bloodstream. Peripheral tolerance mechanisms are critical to silence these cells and act through induction of anergy, deletion, or suppression by regulatory T-cells (T-reg, CD4+Foxp3+). Although many immune cells are involved in this process, dendritic cells (DCs) play a central role since they can promote peripheral tolerance by presenting the antigen to T-cells in steady-state, that is in absence of pro-inflammatory stimuli [22]. T-cell costimulatory pathways also contribute to central and peripheral immunotolerance, with a major role of the Programmed Death 1 (PD-1) and its ligands, Programmed Death Ligand 1 (PD-L1) which is expressed in the thymic cortex, and Programmed Death Ligand 2 (PD-L2) which is expressed in the thymic medulla. Binding of PD-1 to PD-L1/2 inhibits T-cell activation, especially in the case of persistent antigenic stimulation, such as self-antigens encounter, chronic infections or tumors. PD-1 pathway, in fact, contributes to autoimmunity prevention, directly causing T-cell exhaustion and promoting an immunosuppressive tumor microenvironment, partly by T-reg induction [23].

## 3. Immunopathology of TETs

Thymus pivotal role in the immune system development explains the high rate of autoimmune phenomena in TET patients. In fact, approximately 40% of Tm patients suffer systemic symptoms despite being affected by a localized disease, because of the development of autoimmune and paraneoplastic syndromes [24].

The most common paraneoplastic autoimmune syndrome associated with TETs is MG, which occurs in approximately 30% of all Tm patients [25,26]. Nevertheless, MG incidence can vary, depending on the histological subtype: it is approximately 15% in type A Tms, 20% in type AB, 40% in type B1, 50% in type B2, 50% in type B3 and <5% in TCs patients [4]. Furthermore, among Tm patients, 4–7% of those with MG have more than one paraneoplastic syndrome, while approximately 10–15% of those without MG have symptoms from a different paraneoplastic condition [25,26]. MG is caused by the presence of autoantibodies against the nicotinic acetylcholine receptor (AchR) that cause an autoimmune disorder of the postsynaptic portion of the neuromuscular plaque, whose clinical presentation is characterized by fluctuating skeletal muscle weakness and fatigue [27,28].

Tm is also associated with other otherwise rare neurological paraneoplastic syndromes, including: limbic encephalitis, usually due to the presence of anti-Hu antibodies, Ma2 antibodies and CRMP-5 antibodies; neuromyotonia, an autoimmune potassium channelopathy; stiff person syndrome, caused by anti-GAD antibodies; myotonic dystrophy, an inherited disease, occasionally described in association with Tm; polymyositis, an inflammatory myopathy characterized by proximal muscular weakness, frequently associated with MG, myocarditis and autoimmune hepatitis. Non-neurological syndromes may also appear in concomitance with Tm: myocarditis with diffuse myocardium degeneration and infiltration of inflammatory giant cells is generally observed in concomitance with polymyositis and MG; Graft-Versus-Host-Disease, a T-cell mediated condition usually observed after stem cell transplantation, is also rarely seen in patients with Tm; Good syndrome, characterized by hypogammaglobulinemia with immunodeficiency involving both B- and T-cells; pure red cell aplasia, likely caused by T-cell dysfunction. Other rare autoimmune diseases such as thyroiditis, mixed connective tissue disease, paroxysmal nocturnal hematuria, acrokeratosis, systemic lupus erythematosus, pemphigus, lichen planus, conjunctivitis and cancer-associated retinopathy have been reported [25].

In contrast to Tm, TC has more aggressive clinical features with evident local invasiveness, early nodal intrathoracic dissemination and a high propensity to distant metastases, but it is not associated with paraneoplastic syndromes [4,5]. This may be due to the fact that, usually, there are no immature T-cells within TC, so this tumor is rarely associated with autoimmune diseases [29].

Central tolerance and immunoregulation deficiency that occur as a consequence of tumor immune-microenvironment alterations are critical in the pathogenesis of Tm-associated autoimmunity. Indeed, AIRE, AchR and Foxp3 (the transcription factor that induces T-reg phenotype) mRNA levels are lower in tumor tissue from Tm patients without MG compared to those who have developed MG [30]. Moreover, thymopoiesis, that is the maturation of T-cells occurring within the Tm, alters peripheral T-cell repertoire by primarily increasing the proportion of CD45RA+CD8+ T-cells, which is also the population that is reduced after thymectomy [30,31]. Furthermore, autoimmune syndromes can be associated with specific genomic alterations. A higher rate of aneuploidy was observed in Tm from patients with MG. Indeed, the analysis of intratumoral transcripts levels showed that some genes with sequence similarity to major autoimmune targets were overexpressed in Tm associated with MG compared to those without MG. In particular, the medium-sized neurofilament (*NEFM*), whose protein product exhibits immunogenic similarities with the AChR α-subunit and titin, is mainly overexpressed in MG+ A/AB Tms, while the neuronal RYR3, whose encoded protein shares homology with muscular RYR1 and cardiac RYR2, is predominantly overexpressed in MG+ B Tms [16]. Similarly, a differential expression analysis of 34 Tms with or without MG (N = 16 and N = 18, respectively) identified 140 differentially expressed genes [32]. In particular, insulin-like growth factor-binding protein (*IGFBP1*), Krüppel-like factor 15 (*KLF15*) transcription factor, pyruvate dehydrogenase kinase (*PDK4*) and hypoxia-inducible factor (*HIF3A*) were more expressed in Tm associated with MG than in those not associated with it, thus suggesting a role for these genes expression, especially *HIF3A* and *IGBP1*, in the pathogenesis of MG [32]. 

## 4. Implications of TETs Immune Biology for the Clinic: Immunotherapy in TETs

### 4.1. General Considerations about Immunotherapy in TETs

During transformation, cancer cells should escape surveillance by the host immune system in order to proliferate. To do so, the cancer cell abnormally expresses proteins that hamper T-lymphocyte activation despite proper antigen recognition, which are referred to as immune checkpoints [33]. The most biologically and clinically relevant immune checkpoints identified to date are PD-L1 and cytotoxic T-lymphocyte-associated protein 4 (CTLA-4), which are targeted by drugs called “immune checkpoint inhibitors” (ICI). In the last few years, the use of ICI has revolutionized the treatment and the prognosis of many cancers and has shown the ability to reach durable responses in a subset of patients [34,35,36,37,38,39,40,41,42,43,44,45,46,47]. Currently, the identification of patients likely to respond to ICI is based on three predictive factors: PD-L1 expression on tumor cells, microsatellite instability (MSI), and tumor mutational burden (TMB) [48,49,50,51]. While PD-L1 expression and MSI are clinically validated, TMB is used only in investigational studies.

ICIs are characterized by a manageable safety profile and better tolerability to patients compared to standard chemotherapy. Nevertheless, immune checkpoint inhibition induced an all-new class of adverse events, i.e., immune-related adverse events (irAEs), that are caused by the activation of autoreactive T-cells and other immune cells against host tissues, thus mocking an autoimmune disease (AD). irAEs are commonly manageable and mild in severity, but can rarely be fatal, with a toxicity fatality rate ranging from 0.36% to 1.23% depending on the type of ICI used (lowest with anti-PD-1 and anti-PD-L1, followed by anti-CTLA-4, and highest with the combination of anti-PD1/PDL1 and anti-CTLA-4) [52,53]. Skin, thyroid, colon, lung and pituitary gland are the most commonly involved organs, with differences among drugs. Heart, nervous system and other organs can be more rarely affected with more severe toxicities [53,54,55]. In addition, the development of irAEs is associated with improved outcome on ICIs in different cancer types [56,57,58,59]. Patients that are candidates to receive ICIs, which are affected by a preexisting AD or with a predisposition to developing an AD, need a comprehensive and prudent evaluation. In fact, patients with an AD or receiving an immunosuppressive therapy were usually excluded from clinical trials of ICIs, due to the possible risk for an autoimmune flare on ICIs and the reduced efficacy of ICI during immunosuppression. Nonetheless, subsequent experiences suggested that ICIs might be safe in selected patients with a preexisting AD and the onset of autoimmunity flares were manageable (exacerbation rate of 20–40%) [60,61,62]. As for the efficacy, small case series reported complete response in patients treated simultaneously with ICIs and selective immunosuppressant therapy (e.g., tocilizumab, vedolizumab, infliximab) [63,64], while nonselective immunosuppressant drugs (e.g., corticosteroids, tacrolimus, cyclophosphamide, mycophenolate mofetil) negatively affected the prognosis of patients on ICI [65]. However, a worse outcome on steroids has been reported to be dependent on the indication because of which these are prescribed [66].

ICIs have also been investigated in rare cancers, including TETs. PD-L1 is expressed on the normal thymic cortex cells and it is also highly expressed on TETs tumor cells [29]. Tumor-reacting CD8+ T-cells and consequent IFNγ production cause expression of PD-L1 and other immunosuppressive proteins, such as indoleamine 2,3-dioxygenase (IDO1) and Foxp3, on tumor cells as a feedback mechanism, as shown in melanoma cells [67]. In TETs, the abnormal thymic architecture and thymic epithelial cells are responsible for the development of dysfunctional thymocytes and potentially autoreactive T-cells, which are released into the circulation. These autoreactive T-cells may recognize self-antigens expressed on TET tumor cells, causing IFNγ release, which in turn upregulates PD-L1 expression on TET tumor cells. A positive expression of PD-L1, defined as IHC score ≥ 1 or positive staining ≥ 1%, is found in around 70% of TCs and 20% of Tms (types A, AB and B), but its association with the outcome is discordant [29,68,69]. High PD-L1 expression in TETs, defined as an IHC score ≥ 3 or positive staining ≥ 50%, is associated in some studies with more advanced disease stage (stage III-IV in the Masaoka–Koga classification) and more aggressive histology types, such as in type B2/B3 Tm or TC [68,69,70,71]. As opposed to this, high PD-L1 expression was associated with an increased number of infiltrating cytotoxic T-lymphocytes and improved survival in a series of 25 TCs [72]. Regarding other immunosuppressive molecule expression, a retrospective study on surgically resected TETs revealed a high expression of IDO and Foxp3 in 13% and 16% of Tms, respectively, and both associated with high grade tumor histology, but had no survival differences. IDO and Foxp3 were overexpressed in 14% and 29% of TCs, respectively, and were not associated with stage or grade, but a longer survival in patients whose TC had low expression of IDO and high expression of Foxp3 was observed [73]. TETs are overall characterized by a low TMB, which is the lowest among adult cancers, but it is significantly higher in TCs compared to other TETs [16,74,75], while MSI has been only exceptionally described in TCs [16]. The high expression of PD-L1 provides a rationale for the use of immunotherapy in TET, but concerns are risen due to the association of TETs with the aforementioned autoimmune manifestations. In this scenario, TC might be the favorite setting for immune checkpoint inhibition, given its high PD-L1 expression levels and its high TMB, alongside with a lower incidence of autoimmune paraneoplastic syndromes compared to Tm [16]. Currently, three ICIs across four studies have been evaluated in TET patients (Table 1): pembrolizumab (a humanized IgG4 kappa anti-PD1 antibody), nivolumab (a fully human IgG4 anti-PD1 antibody) and avelumab (a fully human IgG1 anti-PD-L1 antibody).

### 4.2. Nivolumab

The PRIMER study is the only trial of nivolumab that investigated activity in TET patients, but it was closed early for futility at the first stage [76]. A total of 15 pretreated patients with unresectable or recurrent TC, regardless of PD-L1 expression, were enrolled in this two-stage single arm phase II trial. In total, 13 TCs (87%) had squamous histology, nine patients (60%) had previously received one or two lines of chemotherapy while six patients (40%) had undergone three or more lines of therapy. Nivolumab 3 mg/kg was administered every 2 weeks until disease progression or unacceptable toxicity. The primary endpoint was ORR. No complete nor partial responses (CR, PR) was observed, 11 patients had stable disease (SD), and four patients had progressive disease (PD) as the best response. The disease control rate (DCR) was 73.3%, the mPFS was 3.8 months (95% CI: 1.9–7.0) and the median overall survival (mOS) was 14.1 months (95% CI: 11.1-Not estimable).

The toxicity profile was overall manageable: there were no drug discontinuations because of AEs, the majority of AEs was mild, and only two patients presented serious irAEs (one grade 3 transaminase increase and one grade 2 adrenal insufficiency, according to Common Terminology Criteria for Adverse Events, version 4.0 grading) [76].

### 4.3. Avelumab

The JAVELIN Solid Tumor study (NCT01772004) is a phase I trial of avelumab in advanced solid tumors that enrolled a total of eight heavily pretreated TET patients (seven with Tm and one with TC) [77]. Patients had no history of Tm-associated autoimmune disease. Avelumab at doses of 10 mg/kg to 20 mg/kg every 2 weeks was administered until disease progression or intolerable toxicity, obtaining disease shrinkage in four patients (two confirmed PR and two unconfirmed PR, all in Tm patients) and stable disease in three patients (two with Tm and one with TC). Remarkably, all responders developed irAEs and had previously been treated with sunitinib, a multikinase inhibitor with an antiangiogenic effect. On the contrary, only one out of the four non-responders developed an irAE. However, avelumab was worse tolerated compared to available data in other solid tumors, and a higher-than-expected rate of severe irAEs was observed. Indeed, the incidence of grade 3 AEs was 38% (N = 3), with the same rate for grade 4 AEs (38%, N = 3). Overall, five patients developed severe irAEs. Specifically, one patient with a B3 Tm developed grade 3 creatine phosphokinase elevation and grade 1 transaminase elevation 2 weeks after the first avelumab administration. Another patient with a B3 Tm developed bulbar weakness, mild sensory loss in feet, facial diplegia, weakness of the tongue and hypophonia after 3 avelumab administrations. The patient obtained only a partial recovery after oral steroid therapy. A patient with a B2 Tm developed myositis after the first administration of avelumab. Another patient with a B2 Tm developed progressive dysphagia and generalized muscle weakness after the first administration of avelumab. This patient required intensive care unit admission with intubation and mechanical ventilation because of progressive dyspnea and dysphagia and was treated with intravenous steroid therapy that led to only partial recovery. Lastly, one patient with a B1 Tm developed grade 3 enteritis with subsequent grade 4 hypokalemia after 11 administrations of avelumab.

Correlative analyses showed that pretreatment absolute lymphocyte count was higher in responders compared to non-responders, while frequencies of B cells, regulatory T-cells, conventional dendritic cells, and natural killer cells were lower in responders compared to non-responders. Additionally, T-cell receptor diversity in pre-therapy peripheral blood mononuclear cells (PBMC) was higher in patients who responded and developed irAEs compared to those who did not. PD-L1 expression could only be analyzed in two tumor samples and thus its association with the response could not be determined in this study. Interestingly, intratumoral immune infiltrates evaluated on two paired biopsies (pre- and post-treatment) showed that the immune pre-treatment infiltrate mainly composed of immature T-cells, shifted towards predominantly mature CD8+ T-cells infiltrate in the patient whose Tm responded to treatment, while it kept an immature T-cells predominance in the patient whose Tm remained stable. In addition, in the patient who achieved a partial response and developed immune-related enteritis, HLA I expression was increased in post-treatment biopsies of the tumor and gastrointestinal tract [77].

### 4.4. Pembrolizumab

Pembrolizumab was tested on patients with recurrent TC in a phase II trial [78]. Forty patients progressing on chemotherapy were included, while patients with a history of autoimmune disease were excluded. The predominant histology was squamous carcinoma (48%), followed by poorly differentiated (32%) and neuroendocrine (15%) carcinoma. Pembrolizumab was given at a 200 mg every 3 weeks dose for up to 2 years or until disease progression or intolerable toxicity. ORR, which was the primary endpoint, was 22.5% with a DCR of 75% (3% of CRs, 20% of PRs and 53% of SDs).

The most common grade 3–4 irAEs were increased aspartate aminotransferase (N = 5, 13%) and alanine aminotransferase (N = 5, 13%), while one or more severe autoimmune toxicity was observed in six patients (15%). In particular, two patients (5%) developed polymyositis and myocarditis after two administrations of pembrolizumab. High-dose steroid therapy until recovery and pacemaker placement were needed for both patients. In one of these patients, an increased frequency of some T-cell receptor clones in post-treatment blood samples was observed.

PD-L1 expression assessment, available in 37 patients, showed a high expression of PD-L1 (expression of at least 50% of the tumor cells) in 27% of TCs. Among patients with high PD-L1 expression, it was reported that five had a PD and six a CR/PR. Among the 27 TCs with low PD-L1 expression (73%), 23 had PD (85%). In a post-hoc analysis, a benefit in terms of PFS and OS was observed in patients whose TC had high PD-L1 expression (median PFS: 24 months vs. 2.9 months-median OS: not reached vs. 15.5 months, in the PD-L1 high vs. PD-L1 low group, respectively).

A NanoString gene expression profiling was performed in 33 patients, revealing a higher expression of an 18-gene T-cell-inflamed interferon-γ gene profile in responders compared to non-responders. Targeted exome sequencing was also conducted in 36 patients. There was a median of 3 mutations per patient (range 0–12) and *TP53* was the most commonly mutated gene (N = 13, 36%). *TP53* mutations were more frequent in patients with low or no PD-L1 expression, while *CYLD* mutation (N = 5) was associated with high PD-L1 expression. A post-hoc analysis showed a correlation between *TP53* mutation and shorter OS. There was also a non-significant trend between *CYLD* mutation and longer PFS and OS [78]. *GTF2I* mutation, a common mutation in Tm but rare in TC [79], was observed in a patient who developed a severe irAE [78].

Another phase II trial of pembrolizumab enrolled 33 TET patients (26 with TC and 7 with Tm) after progression to at least one line of platinum-based chemotherapy [80]. Pembrolizumab 200 mg was given every 3 weeks for 2 years or until disease progression or intolerable toxicity. Patients with an autoimmune disease requiring systemic treatment within the past year or with a history of severe autoimmune disease were excluded. A total of 19 patients (57.3%) had received at least two prior lines of chemotherapy, while 10 (30.3%) had received three or more lines of chemotherapy. In total, 19 of 26 patients with TC had squamous cell carcinoma. The primary endpoint was ORR. Among Tms (N = 7), 2 PRs (28.6%) and 5 SDs (71.6%) were observed, with an ORR of 28.6%, a DCR of 100% and a median duration of response (mDOR) that was not reached. In the TC group (26 patients), five patients had a PR (19.2%) and 14 patients a SD (53.8%), with an ORR of 19.2%, a DCR of 73.1% and a mDOR of 9.7 months. After a median follow-up of 14.9 months, the median PFS was 6.1 months for both Tm (95% CI: 4.3–7.9) and TC patients (95% CI: 5.1–7.1) and the median OS was 14.5 months for TC and not reached for Tm patients.

Grade 3–4 irAEs were observed in 71.4% of Tm patients, the most common being myocarditis (N = 3), hepatitis (N = 2), thyroiditis (N = 1), colitis (N = 1) and nephritis (N = 1). Among patients with TC, grade 3–4 irAEs occurred in 15.4% of cases: hepatitis (N = 2), MG (N = 2) and subacute myoclonus (N = 1). A patient with a B2 Tm treated with immunosuppressant therapy for a grade 4 autoimmune hepatitis, a grade 3 colitis and a grade 2 dermatitis, died because of superimposed cytomegalovirus infection. Three patients with pre-existing AD in their history developed severe irAEs. Considering the alarming incidence of severe irAEs, a study protocol amendment was emanated to exclude from the study patients with Tms and patients with a history of AD.

PD-L1 expression was evaluable in 24 patients and high expression (≥50% tumor proportion score) was observed in 14 tumors. Among tumors with high PD-L1 expression, five patients achieved a partial response while there was no response in the group with low PD-L1 expression (*p* = 0.034). There was no correlation between irAEs and high PD-L1 expression. Similarly, PD-L1 mRNA expression assessed using the nCounterPanCancer Immune Profiling Panel was higher in the four patients who responded compared to those who did not [80].

## 5. Balancing between Efficacy and Toxicity of ICIs in TETs

Immunotherapy showed an interesting antitumor activity in advanced pretreated TETs, partly comparable to what observed in other solid tumors [36,42,47,76,77,78,80], so that pembrolizumab has been added as a second-line option for the treatment of patients with TC by the National Cancer Comprehensive Network (NCCN) guidelines [7]. Similar to what was observed in other cancers, a correlation between high PD-L1 expression and response to PD-1 blockade has been observed in TETs [78,80].

Nevertheless, ICI studies in TET patients reported high rates of grade 3–4 irAEs, ranging from 13% to 38%, even in patients with no previous history of AD or concurrent immune-related paraneoplastic syndrome. Interestingly, the development of irAEs has been associated with the response, as described in other malignancies, including non-small cell lung cancer and melanoma [56,58,77,78,81]. Most commonly reported grade 3–4 irAEs in TET trials were increased liver transaminases, myalgia, myositis, enteritis, myocarditis, thyroiditis, colitis and nephritis [76,77,78,80]. Multiple coexisting autoimmune adverse events were reported in many patients and required medical therapy and hospitalization, with fatal outcome for cardiac toxicity [55,82]. In one of the available phase II trials of pembrolizumab that enrolled both Tm and TC patients, severe adverse events were more commonly reported in Tm compared to TC patients (71.4% vs. 15.4%, respectively) [80].

irAEs reported in TETs trials, such as musculoskeletal, neuromuscular and cardiac irAEs, are usually less common in other cancer types treated with the same agents, for a reported incidence of less than 1% [53,54,55]. This can be due to the thymus function and the unique predisposition to immune paraneoplastic syndromes observed in TET patients, especially Tm ones [16,83]. Since PD-1/PD-L1 interaction plays a crucial role in the normal development of immune tolerance in peripheral lymphoid organs, it is therefore possible that ICIs could alter the mechanism of thymic epithelial cell death, cause loss of immune tolerance and eventually result in the development of a high rate of irAE [23].

In light of this, biomarkers are needed in order to identify both patients who are more likely to benefit from immunotherapy and those who are at risk of developing severe irAEs. Several potential response biomarkers have been evaluated, including PD-L1 expression on tumor cells, but also peripheral blood immune cells subsets, TCR sequencing, tumor immune infiltrate, HLA expression, expression profiles of genes involved in inflammation and specific genetic mutations [77,78,80]. Currently, PD-L1 expression is the only clinically validated predictive biomarker for ICIs targeting the PD-1/PD-L1 axis. PD-L1 expression on tumor cells is finely tuned at different levels by means of tumor intrinsic and extrinsic factors [84,85,86]. The role of *CYLD* in regulating PD-L1 expression has been recently investigated in TETs. *CYLD*, a tumor suppressor gene mutated in patients with familial cylindromatosis and in more than 10% of TCs, is a negative regulator of inflammation. Its loss of function determines an increase in PD-L1 expression, mediated by INF-γ through the activation of the STAT1/IRF1 pathway [87]. Low expression of *CYLD* is associated with high PD-L1 expression, as observed also in the trial of pembrolizumab for TCs [78,87].

On the other hand, predictors for toxicity still need to be identified: histology (Tm vs. TC), history of AD, PD-L1 expression, sex, ethnicity, mutational profile, TCR sequencing and HLA expression may have a role in predicting irAEs development on ICI [77,78,80].

Furthermore, patients with pre-existing ADs are at high-risk of autoimmune flares [60,62]. This theoretically applies also to patients with immune-mediated paraneoplastic syndromes, such as TET ones. In this setting, strategies aimed at preventing ADs or immune-mediated paraneoplastic syndromes exacerbations as well as the development of severe irAEs in high-risk patients, like TET ones, are needed. However, to date no study or guideline addresses these issues. A multidisciplinary and personalized approach is thus necessary. As selective immunosuppressant seems not to affect ICI effectiveness while controlling autoimmune flares in patients with pre-existing ADs, this can be an option to be investigated, at least in high-risk patients, like TET patients, especially those with Tm [60].

## 6. Future Perspectives: Ongoing Combination Trials of ICIs and Tyrosine Kinase Inhibitors (TKIs)

As for other malignancies, there are high expectations in drugs under development for TETs, especially TKIs, ICIs and the combination of both, which are being investigated in many ongoing trials (Table 2). 

Three phase II trials are currently ongoing to investigate in pre-treated patients the safety and clinical activity of avelumab in Tm or TC patients (NCT03076554), nivolumab in B3 Tm and TC patients (The NIVOTHYM trial, NCT03134118) and atezolizumab in TC patients (NCT04321330). Data from these trials of ICIs monotherapy are awaited to clarify the feasibility and the utility of ICI in TETs.

Combining ICI with ICIs or other immune-directed agents is an appealing strategy to increase the rate of responders and to improve outcomes. IDO1 is an intracellular enzyme that catalyzes the first and rate-limiting step of the tryptophan–kynurenine metabolism pathway, causing the cell-cycle arrest, effector T-cell apoptosis and determining local immunosuppression. IDO1 activation has been correlated with poor prognosis in several cancers [88]. The phase II trial of pembrolizumab in advanced TC (NCT02364076) has been amended to include an additional cohort of patients treated with pembrolizumab and epacadostat (an IDO1 inhibitor) on the basis of initial promising data in patients with melanoma and NSCLC [78]. Results are not available yet and recruitment has been halted after the failure of this combination in a large randomized phase III trial for advanced melanoma. 

SO-C101 is a superagonist fusion protein of interleukin-15 (IL-15) and the IL-15 receptor α (IL-15Rα) that had shown to stimulate proliferation, differentiation and anti-tumor action of natural killers (NK) and CD8+ T-cells [89]. The phase I trial NCT04234113 is investigating the safety and tolerability of SO-C101 as monotherapy or in combination with pembrolizumab in patients with pre-treated advanced solid tumors, including TETs.

Targeted therapy with TKI is a promising treatment for TETs, given good efficacy and the manageable safety profile. As TKIs are characterized by a shorter time to response and a higher response rate compared to ICIs, ICIs show a prolonged benefit in those who respond to treatment [77,78,80,90,91]. The combination of an ICI and a TKI is an appealing strategy that could provide patients the benefits of both strategies, but also increase toxicity. The phase II trial NCT03463460, testing the combination of pembrolizumab and sunitinib is currently ongoing in refractory metastatic or unresectable TC patients. The primary endpoint of the trial is the activity of the treatment measured by ORR, while the secondary endpoints are the incidence of irAEs, PFS and OS. Furthermore, the study aims at exploring a possible correlation between clinical benefit and PD-L1 expression level and to investigate whether sunitinib could affect PD-L1 expression, and the number of tumor-infiltrating lymphocytes and of myeloid-derived suppressor cells in the tumor and peripheral blood. Another phase I/II trial (NCT03583086) is a dose escalation and dose expansion study of vorolanib, an oral VEGFR/PDGFR kinase inhibitor, in combination with nivolumab in patients with refractory thoracic malignancies, including TC patients. The endpoint for the phase I part of the study is assessing the safety and tolerability of the combination, while the primary endpoint of the phase II part is to evaluate the efficacy, by measuring ORR. In addition, an explorative analysis will assess the correlation of PD-L1 expression and TMB with the experimental treatment. Finally, the preliminary results of the phase II CAVEATT study have been recently published: 1 patient with B3 Tm and 12 with TCs in progression after at least one line of platinum-based chemotherapy were enrolled and treated with avelumab in combination with axitinib, an oral VEGFR-1/2/3 kinase inhibitor. 40% of patients had a PR, 60% a SD with a mPFS of 7.9 months. G3–4 AEs were reported in two patients and were attributable to axitinib (hypertension and hand-foot syndrome), while no irAEs were described. The trial accrual is currently ongoing [92].

## 7. Conclusions

The thymus has a unique role in the development of adaptive immunity fostering the maturation of T-cells through the proper selection of non-self-reactive clones. Due to this, treatment with ICIs is associated with a high rate of irAEs in TET patients, so that, to date, chemotherapy still represents the backbone of systemic treatment for these patients. Available data do not support the use of ICIs in TET patients outside of clinical trials, especially in patients with Tms and autoimmune paraneoplastic syndromes, because of the high risk for severe irAEs. However, TC showed higher response rates with lower rates of irAEs compared to Tms. The latter is however still higher than what was observed in other solid tumors. The identification of factors to predict response and/or the development of irAEs in TET patients (e.g., Tm vs. TC, PD-L1 expression, TMB, HLA, PBMC and T-cell receptor density among others) could help in weighing risks and benefits and guide the choice of active treatments (ICI vs. chemotherapy/TKI) and prevention immunosuppressive strategies, once validated.

## Figures and Tables

**Figure 1 ijms-21-09056-f001:**
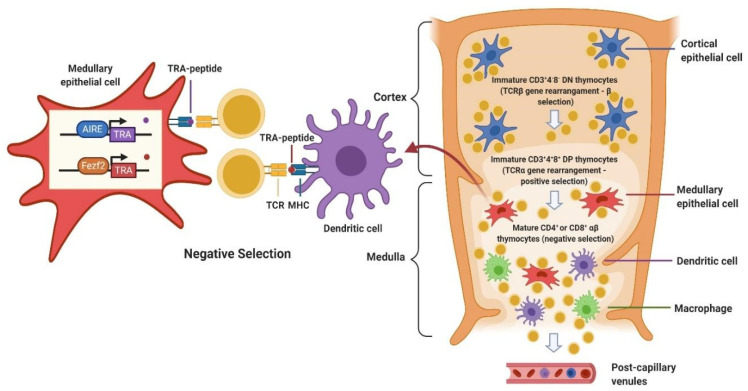
T-cell development process. DN: double negative; DP: double positive; MHC: major histocompatibility complex; TCR: T-cell receptor; AIRE: Auto Immune Regulator; TRA; tissue-restricted antigen; Fezf2: forebrain embryonic zinc finger-like protein 2.

**Table 1 ijms-21-09056-t001:** Published clinical trials of IO monotherapy in thymic epithelial tumor (TET) patients.

First Author, Year (Study Name)	Phase	TC (n)	Tm (n)	Experimental Drug	mPFS, % (95% CI)	ORR, % (95% CI)	G3–G4 irAEs n (%)
Giaccone, G., et al., 2018 (NCT02364076)	II	40	0	Pembrolizumab	4.2 months (2.9–10.3)	22.5% (10.8–38.5)	6 (15%)
Cho, J., et al., 2019 (NCT02607631)	II	26	7	Pembrolizumab	6.1 months (5.3–6.9)	21.2% (10.7–37.8)	9 (27.3%)
Katsuya, Y., et al., 2019 PRIMER study (NCCH1505)	II	15	0	Nivolumab	3.8 months (1.9–7.0)	0% (0–21.8)	2 (13.3%)
Rajan, A., et al., 2019 JAVELIN (NCT01772004)	I	1	7	Avelumab	NA	57% (NA)	5 (62.5%)

IO: immuno-oncology; TC: thymic carcinoma; Tm: thymoma; mPFS: median progression free survival; CI: confidence interval; ORR: overall response rate; irAEs: immune-related adverse events; NA: not available.

**Table 2 ijms-21-09056-t002:** Active and recruiting clinical trials of immune checkpoint inhibitor (ICI) monotherapy and combinations (source: clinicaltrials.gov; last accessed: 26 October 2020).

Trial (NCT)	Phase	Disease	Setting	Experimental Drug	Estimated Enrolment	Primary Endpoint
NCT03076554	II	TC and Tm	Pre-treated with Platinum-based CHT	Avelumab	55	ORR, safety
NCT03134118	II	TC and Tm (B3)	Pre-treated with Platinum-based CHT	Nivolumab	55	PFS at 6 months
NCT04321330	II	TC	Pre-treated	Atezolizumab	34	ORR
NCT04234113	I/Ib	Solid tumors including TETs	Pre-treated	SO-C101 ± pembrolizumab	96	Safety
NCT03463460	II	TC	Pre-treated with Platinum-based CHT	Pembrolizumab and Sunitinib	40	ORR
NCT03583086	I/II	Thoracic tumors including TC	Pre-treated	Vorolanib and nivolumab	177	Safety and ORR
2017-004048-38 (CAVEATT study) *	II	TC and Tm (B3)	Pre-treated with Platinum-based CHT	Avelumab and Axitinib	33	ORR

IO: immuno-oncology; TC: thymic carcinoma; Tm: thymoma; CHT: chemotherapy, ORR: overall response rate. * EUDRACT trial registration number.

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
