# Peer review of "Immunobiology of Thymic Epithelial Tumors: Implications for Immunotherapy with Immune Checkpoint Inhibitors"

_ijms, 2020, doi:10.3390/ijms21239056_

Round 1

Reviewer 1 Report

The manuscript by Tateo et al. provides a comprehensive review on the biology and immune-mediated therapies of thymic epithelial tumors.

Author Response

The manuscript has been thoroughly revised with grammatical and syntactical changes. The authors would like to thank you for the time spent reading and commenting on the manuscript. Thank you for your contribution. 

Reviewer 2 Report

The manuscript focused on the immunobiology and immunotherapy with immune checkpoint inhibitors of thymic epithelial tumors, covering a brief introduction on the clinical and pathologic classification, the process of T cell maturation, the latest result of immune checkpoint inhibitor clinical trial, and future perspectives derived from current studies on the immunopathology of thymic epithelial tumors. The manuscript is well organized and concisely written. As a reviewer for the manuscript, I would like to add certain perspectives and references so that the scope would be completed.

  1. In the section of 【Immunopathology of TETs】:

Tumor-microenvironment alterations and mRNA levels of certain factors such as AIRE, AchR, and Foxp3 have been reported to associate with thymoma-associated autoimmunity. It would be better to include the main findings of "Molecular profiling of thymoma with myasthenia gravis: Risk factors of developing myasthenia gravis in thymoma patients" In Lung Cancer 139: 157-164, that “In thymoma with MG, HIF3A, IGFBP1, KLF15, and PDK4 were significantly up-regulated comparing to non-MG thymoma. The expression of HIF3A in thymoma was involved in the development of MG.”

  1. In section 【1 General considerations about immunotherapy in TETs】:

The fact that TETs lack normal thymic architecture and have abnormal thymic epithelial cells is responsible for the development dysfunctional thymocytes and autoreactive T cells released into the circulation. The autoreactive T cells recognize self-antigens expressed on TET tumor cells and release IFN-gamma, which in turn upregulate PD-L1 expression in TET tumor cells. These findings provide an explanation for high tumor cell PD-L1 expression in TETs. Although most studies showed high PD-L1 expression associated with aggressive histology, certain contradictory results exist among different studies, especially in “"Prognostic Value of Programmed Death Ligand 1 and Programmed Death 1 Expression in Thymic Carcinoma" in Clin Cancer Res 22(18): 4727-4734.

  1. Also, the role of Treg Foxp3 in the pathogenesis and IDO as a potential target have been described. The interesting findings in “"Different pattern of PD-L1, IDO, and FOXP3 Tregs expression with survival in thymoma and thymic carcinoma" in Lung Cancer 125: 35-42 should be included.

  1. On page 8, the sentence “Grade 3-4 irAEs were observed in 71.4% of Tm patients and were…” was NOT complete. Also, in the last paragraph on page 8, “…【it】 is therefore possible that ICIs…”, please revise it.

Author Response

The authors would like to thank the reviewer for these comments.

The manuscript focused on the immunobiology and immunotherapy with immune checkpoint inhibitors of thymic epithelial tumors, covering a brief introduction on the clinical and pathologic classification, the process of T cell maturation, the latest result of immune checkpoint inhibitor clinical trial, and future perspectives derived from current studies on the immunopathology of thymic epithelial tumors. The manuscript is well organized and concisely written. As a reviewer for the manuscript, I would like to add certain perspectives and references so that the scope would be completed.

 In the section of 【Immunopathology of TETs】:

Tumor-microenvironment alterations and mRNA levels of certain factors such as AIRE, AchR, and Foxp3 have been reported to associate with thymoma-associated autoimmunity. It would be better to include the main findings of "Molecular profiling of thymoma with myasthenia gravis: Risk factors of developing myasthenia gravis in thymoma patients" In Lung Cancer 139: 157-164, that “In thymoma with MG, HIF3A, IGFBP1, KLF15, and PDK4 were significantly up-regulated comparing to non-MG thymoma. The expression of HIF3A in thymoma was involved in the development of MG.”

Lines 159-164: insertion of a paragraph with its reference as suggested in observation n°1.

“Similarly, a differential expression analysis of 34 Tms with or without MG (N=16 and N=18, respectively) identified 140 differentially expressed genes [32]. In particular, insulin-like growth factor-binding protein (IGFBP1), Krüppel-like factor 15 (KLF15) transcription factor, pyruvate dehydrogenase kinase (PDK4) and hypoxia-inducible factor (HIF3A) were more expressed in Tm associated with MG than in those not associated with it, thus suggesting a role for these genes expression, especially HIF3A and IGBP1, in the pathogenesis of MG [32].”

  1. In section 【1 General considerations about immunotherapy in TETs】:

The fact that TETs lack normal thymic architecture and have abnormal thymic epithelial cells is responsible for the development dysfunctional thymocytes and autoreactive T cells released into the circulation. The autoreactive T cells recognize self-antigens expressed on TET tumor cells and release IFN-gamma, which in turn upregulate PD-L1 expression in TET tumor cells. These findings provide an explanation for high tumor cell PD-L1 expression in TETs. Although most studies showed high PD-L1 expression associated with aggressive histology, certain contradictory results exist among different studies, especially in “"Prognostic Value of Programmed Death Ligand 1 and Programmed Death 1 Expression in Thymic Carcinoma" in Clin Cancer Res 22(18): 4727-4734.

Lines 202-209: insertion of a paragraph with its reference as suggested in observation n°2.

“Tumor-reacting CD8+ T-cells and consequent IFNγ production cause expression of PD-L1 and other immunosuppressive proteins, such as indoleamine 2,3-dioxygenase (IDO1) and Foxp3, on tumor cells as a feedback mechanism, as shown in melanoma cells [68]. In TETs, the abnormal thymic architecture and thymic epithelial cells are responsible for the development of dysfunctional thymocytes and potentially autoreactive T cells, which are released into the circulation. These autoreactive T cells may recognize self-antigens expressed on TET tumor cells, causing IFNγ release, which in turn upregulate PD-L1 expression on TET tumor cells.”

Lines 214-215: insertion of a paragraph with its reference as suggested in observation n°2.

“As opposite to this, high PD-L1 expression was associated with increased number of infiltrating cytotoxic T-lymphocytes and improved survival in a series of 25 TCs [73].”

  1. Also, the role of Treg Foxp3 in the pathogenesis and IDO as a potential target have been described. The interesting findings in “"Different pattern of PD-L1, IDO, and FOXP3 Tregs expression with survival in thymoma and thymic carcinoma" in Lung Cancer 125: 35-42should be included.

Lines 215-220: insertion of a paragraph with its reference as suggested in observation n°3.

“Regarding other immunosuppressive molecule expression, a retrospective study on surgically resected TETs revealed a high expression of IDO and Foxp3 in 13% and 16% of Tms, respectively, both associated with high grade tumor histology, but no survival differences. IDO and Foxp3 were overexpressed in 14% and 29% of TCs, respectively, and were not associated with stage or grade, but a longer survival in patients whose TC had low expression of IDO and high expression of Foxp3 was observed [74].”

  1. On page 8, the sentence “Grade 3-4 irAEs were observed in 71.4% of Tm patients and were…” was NOT complete. Also, in the last paragraph on page 8, “…【it】 is therefore possible that ICIs…”, please revise it.

Lines 326-327: "Grade 3-4 irAEs were observed in 71.4% of Tm patients, the most common being myocarditis (N=3), hepatitis (N=2), thyroiditis (N=1), colitis (N=1) and nephritis (N=1)."

Lines 361-365: "Since PD-1/PD-L1 interaction plays a crucial role in the normal development of immune tolerance in peripheral lymphoid organs, it is therefore possible that ICIs could alter the mechanism of thymic epithelial cell death, cause loss of immune tolerance and eventually result in the development of a high rate of irAE [23]."